# Antioxidant Actions of Melatonin: A Systematic Review of Animal Studies

**DOI:** 10.3390/antiox13040439

**Published:** 2024-04-07

**Authors:** Karla Krislane Alves Costa Monteiro, Marcos Eiji Shiroma, Luciana Lamarão Damous, Manuel de Jesus Simões, Ricardo dos Santos Simões, José Cipolla-Neto, Edmund Chada Baracat, Jose Maria Soares-Jr.

**Affiliations:** 1Laboratório de Ginecologia Estrutural e Molecular (LIM-58), Disciplina de Ginecologia, Departamento de Obstetrícia e Ginecologia, Hospital das Clínicas HC-FMUSP, Faculdade de Medicina, Universidade de São Paulo, São Paulo 05403-010, Brazillucianadamous@gmail.com (L.L.D.); manuel.simoes@fm.usp.br (M.d.J.S.); ricardo.simoes@fm.usp.br (R.d.S.S.); ecbaracat@usp.br (E.C.B.); jose.msjunior@fm.usp.br (J.M.S.-J.); 2Laboratório de Neurobiologia, Departamento de Fisiologia, Instituto de Ciências Médicas (ICB-I), Universidade de São Paulo, São Paulo 05508-000, Brazil; cipolla@icb.usp.br

**Keywords:** melatonin, antioxidant, oxidative stress, nitrosative stress, superoxide dismutase

## Abstract

Melatonin is an indoleamine with crucial antioxidant properties that are used to combat inflammatory and neoplastic processes, as well as control transplants. However, the clinical applications of melatonin have not yet been fully consolidated in the literature and require in-depth analysis. Objectives: This study reviewed the literature on the antioxidant properties of melatonin in rat models. Methods: We followed the guidelines of the Preferred Reporting Items for Systematic Reviews and Meta-analyses and used the PubMed, LILACS, and Cochrane databases, Google Scholar, and article references, irrespective of publication time. Results: Ten articles involving 485 rats were selected, and the effects of melatonin on antioxidant markers were investigated. Melatonin increased superoxide dismutase in nine studies, glutathione peroxidase in seven studies, and catalase in five studies. In contrast, melatonin reduced glutathione in three studies and malonaldehyde in seven of eight studies. Conclusion: Our findings suggest that melatonin effectively reduces oxidative stress.

## 1. Introduction

Melatonin is an amphiphilic indoleamine (N-acetyl-5-methoxytryptamine), is present in several tissues in the body, and has been the target molecule of a substantial amount of recent research. Melatonin production starts from tryptophan, an essential amino acid, which is transformed into 5-hydroxytryptophan by tryptophan hydroxylase. Then, 5-hydroxytryptophan is converted into serotonin, which arylalkylamine N-acetyltransferase acetylates into N-acetylserotonin. Finally, N-acetylserotonin is then converted into melatonin by acetylserotonin O-methyltransferase. Melatonin may be produced in several tissues and organs (e.g., the ovaries, retinas, bone marrow, intestines, placenta, lymphocytes, and liver), where it acts locally through autocrine and paracrine mechanisms. In the pineal gland, melatonin already has the characteristics of a hormone and is produced in accordance with circadian rhythms during the night. Therefore, the duration of melatonin production is bigger during winter when night is longer than in summer [1,2,3].

Considering its strict association with the photoperiod, melatonin is a time signal for the human body because it triggers the physiological mechanisms of adaptation to night, day, and the seasons. Therefore, melatonin regulates several physiological functions, playing essential roles in regulating body temperature, metabolic control, glycemic and lipid control, cardiovascular and immune system control, reproduction and pregnancy—such as the secretion of gonadotropin-releasing hormone, gonadotropins, follicle-stimulating hormone, and luteinizing hormone neural development, and protection [2,3,4].

Melatonin has a chronobiotic role and can regulate circadian rhythmicity. Its immediate effects result from the classic hormonal mode of action related to the presence of melatonin in biological fluids and its interaction with corresponding molecular effects. These effects occur nightly when the pineal gland releases melatonin, which is present in the blood and cerebrospinal fluid. These effects depend on factors such as melatonin receptors, local hormone concentration, type of cellular receptors, and signaling systems [3], according to the phase of the circadian cycle. 

In addition, melatonin is a potent free radical scavenger and an antioxidant targeted at the mitochondria, thereby protecting against oxidative stress and locally reducing reactive oxygen species (ROS) and reactive nitrogen species (RNS) [3,4,5,6,7,8,9,10,11,12,13,14,15,16,17,18,19,20,21]. Melatonin can also neutralize free radicals, diminish the effects of injuries induced by ischemia–reperfusion, and exert neuroprotective and reproductive effects, thus improving oocyte quality [3,4,10,12,13,14,15,16,17,18,19,20,21].

Free radicals are generated during energy metabolism in cells. The accumulated free radicals can lead to extensive cell death should the host have insufficient defense mechanisms to eliminate their harmful effects; thus, a balance is essential to maintain homeostasis. Melatonin and its metabolites have extremely broad antioxidant actions, including the ability to neutralize superoxide anions (O_2_^•−^), hydroxyl radicals (^•^OH), single oxygen (1O_2_), hydrogen peroxide (H_2_O_2_), hypochlorous acid (HOCl), nitric oxide (NO), and peroxynitrite anions (ONOO^−^). The primary product of the reaction of melatonin and other oxidants is cyclic 3-hydroxymelatonin (C_3_OHM). During its action, many melatonin metabolites can act in sequence or together to neutralize ROS/RNS, known as a cascade of the elimination of free radicals [2,3,5,6,7,8], which involves the antioxidant activity of melatonin and its generated metabolites.

Numerous pathways for the formation of N1-acetyl-N2-formyl-5-methoxykynuramine (AFMK) have been identified in vitro and in vivo, and increased levels of melatonin metabolites, AFMK, and N-acetyl-5-methoxykynuramine (AMK) positively regulate antioxidant enzymes and negatively regulate pro-oxidative and pro-inflammatory enzymes [22]. Furthermore, melatonin and its metabolites protect lipids, proteins, and DNA against oxidative damage and nitrosative stress [2,3]. Melatonin inhibits cell apoptosis in some cases, such as in ischemia reperfusion injury, which encompasses cytotoxic mechanisms and includes an inflammatory immune reaction, precipitating damage by oxidative stress and leukocyte mobilization. Furthermore, melatonin has been used in several tissues to reduce the calcium uptake by the cells and modulate the expression of antioxidants [4,8,22,23,24,25,26,27]. Recently, 6-hydroxymelatonin was seen to inhibit cytochrome C release from the mitochondria, suppress caspase 3 activity, and stabilize the mitochondrial membrane potential, thereby protecting against the neuronal cell death induced by oxygen–glucose–serum deprivation in cultured cells [26,27,28,29]. This finding demonstrated the mechanisms of the melatonin antioxidant cascade.

The actions of melatonin go beyond its direct antioxidant actions as a free radical scavenger. It takes indirect actions in stimulating the cellular antioxidant defense system, increasing mRNA and glutathione reductase (GSH) levels, and the activities of several antioxidant enzymes, such as glutathione peroxidase (GPx), catalase (CAT), and superoxide dismutase (SOD), which catalyzes the conversion of O_2_ into H_2_O_2_ [1,3,12,13,14,15,16,17,18,19,20,21]. Evaluating these enzymes aids in verifying the antioxidant effects of melatonin in cells. Using melatonin has also been associated with decreased levels of malondialdehyde (MDA), the stable end product of lipid peroxidation and an indirect indicator of increased intracellular ROS generation. Consequently, using melatonin influences a decrease in lipid and peroxidation levels and reduces cell death [15].

However, the clinical applications of melatonin are still evolving, and elucidating its physiological and clinical effects remains challenging. Therefore, we reviewed the scientific literature using studies that evaluated the antioxidant effects of melatonin through SOD, CAT, GPx, GSH, and MDA levels. This study aimed to clarify and contribute to the understanding of the antioxidant actions of melatonin in experimental studies and provide a basis for future research.

## 2. Materials and Methods

This systematic review was conducted in accordance with the Preferred Reporting Items for Systematic Reviews and Meta-Analyses (PRISMA) [30]. 

Inclusion criteria: Studies that directly assessed the effects of melatonin on oxidative stress, performed rat experiments using melatonin, and assessed antioxidants written in English, Portuguese, Spanish, French, or Italian. No period restrictions were noted.

Exclusion: Literature reviews, studies using animals other than rats, cell culture and tissue studies, and articles in Japanese, Chinese, German, Polish, and Russian were excluded due to variations in the translation and interpretation of the results.

The study selection was conducted in the PubMed, LILACS, Cochrane databases, Google Scholar, and Science Direct databases, as well as in the references of the articles included in this review, systematic reviews, and meta-analyses. The materials were assessed according to the eligibility criteria at the last visit on 14 September 2023. The search strategy involved using Medical Subject Headings terms (Table 1). 

The selection phase was divided into two distinct stages: the initial screening with the reading of titles and abstracts and the final selection with the identification of rat experiments that used melatonin and assessed the effects of its antioxidant actions in situations of high oxidative stress against a control group. Two independent reviewers (K.K.A.C.M. and J.M.S.J.) carried out these stages. In the event of persistent disagreement, a third reviewer (R.S.S.) was introduced. Studies that met the inclusion criteria were selected for complete reading and in-depth analysis of the results.

After completely reading the articles, data were extracted from each publication and analyzed. A table was drawn with details about the sample size, melatonin dose, and route utilized and an assessment of the oxidative stress tests performed and their results. The analysis complied with the PRISMA 2020 guidelines [30].

MDA, SOD, GPx, GSH, and CAT were mostly used as the antioxidant markers to evaluate antioxidant actions. The results showed diminished oxidative stress, reduced cell and tissue damage, and anti-apoptotic activity. These outcomes are similar to the positive results for melatonin reported in previous reviews.

The risk of bias in the reviewed studies was assessed using Cochrane’s risk of bias test and the Systematic Review Center for Laboratory Animal Experiments. The domain questions included selection, performance, detection, friction, and reporting biases, which were concentrated in the study design. These values are presented in Figure 1 [12,13,14,15,16,17,18,19,20,21,31]. Two reviewers independently evaluated the risk of bias. In the event of discrepancies, a third reviewer was consulted. The judgment of the risk of bias (low, few issues, or high) is displayed for each specific result [31].

## 3. Results

Overall, 991 records were identified, of which 289 were initially removed using the Rayyan’s automatic technique to detect duplicates. Finally, 26 studies were selected to be read fully based on their titles and abstracts. After reading and applying the eligibility criteria, 10 studies were included. The other 16 studies were excluded because the animals were not rats (*n* = 4), the studies involved cell culture and tissues (*n* = 6), the studies did not assess antioxidant markers (*n* = 5), and one text was not available in its entirety (Figure 2) [12,13,14,15,16,17,18,19,20,21].

The 10 studies that were included assessed 485 rats, which were subjected to different experiments: (a) obstructive jaundice induced according to main bile duct ligation and sectioning (*n* = 21); (b) tartrazine-induced neurotoxicity (*n* = 40); (c) valproic acid (VPA)-induced hippocampal neurogenesis (*n* = 48); (d) streptozocin-induced diabetes mellitus (*n* = 104); (e) ischemic preconditioning in renal ischemia–reperfusion injury (*n* = 48); (f) lead-induced male gonadotoxicity (*n* = 50); (g) potassium-dichromate-induced male gonadotoxicity (*n* = 40); (h) liver failure after ischemia–reperfusion injury (*n* = 60); and (i) secondary biliary cirrhosis owing to ligation of the main bile duct (*n* = 24) [12,13,14,15,16,17,18,19,20,21].

The studies administered melatonin doses ranging from 0.1 to 10 mg/kg/day via the oral or intraperitoneal route to quantify oxidative stress. The studies assessed the SOD, CAT, GPx, GSH, and MDA levels in different experimental clinical situations in rats to investigate whether the oxidative reaction increased or decreased. The analyses were performed according to the specific experimental structure of each study, resulting in the absence of a uniform standard for evaluation of the substances (Table 2). 

The experimental group was compared to the control group, and both groups experienced oxidative stress. The antioxidant reaction was greatest in the group that received melatonin, independent of the dose or route. Most of the markers in all the studies increased slightly or strongly, including SOD in nine studies, GPx in seven studies, CAT in five studies, and GSH in three studies, and MDA decreased in seven of eight studies. Two studies also showed the anti-apoptotic effect of melatonin, including a reduction in B-cell lymphoma 2 (Bcl-2) expression and caspase-3 activity [12,13,14,15,16,17,18,19,20,21].

Melatonin stimulates antioxidant enzymes to reduce oxidative damage, suggesting an improvement in tissue quality and the immune system by increasing antioxidant enzymes and mitigating lipid peroxidation by decreasing the MDA levels.

The risk of bias scores ranged from 6.5 to 7, indicating a low risk of bias. However, the lack of data and blinded outcome assessors are issues that could indicate performance and detection biases [30].

## 4. Discussion

Melatonin can stimulate or suppress certain substances involved in the antioxidant process in different tissues. Many mechanisms of action may act through independent receptor signaling (Figure 3). Melatonin’s proprieties are related to oxidative system stability, intracellular cascade signaling (melatonin metabolites), inflammatory systems, and apoptosis. SOD, CAT, GPx (all of which enhanced the melatonin levels), and MDA (which decreased melatonin the levels) are the primary substances involved in melatonin mechanism [1,2,3,4,5,6,7,8,12,13,14,15,16,17,18,19,20,21].

SOD, CAT, and GPx play essential roles in balancing oxidative and antioxidant systems, as do non-enzymatic substances, mainly vitamins (C, E) and trace elements (copper, zinc, and selenium). Herein, the effect of melatonin as an antioxidant is proven by the increase in SOD, CAT and GP_X_ and the decrease in MDA. However, the mechanism of action of melatonin remains unclear. A possible mechanism is its amphiphilic properties (hydrophilic and lipophilic), which facilitate its diffusion into various subcellular compartments, such as membranes, the cytoplasm, the nucleus, and mitochondria, thereby expanding and facilitating its action in various locations. However, this action differs from that of other antioxidants, such as vitamin C (hydrophilic) and vitamin E (lipophilic), which have limited actions due to their physicochemical characteristics [1,3,12,13,14,15,16,17,18,19,20,21].

Furthermore, the antioxidant action of melatonin triggers the melatonin antioxidant cascade effect (independent melatonin receptor signaling), where the first, second, and third-generation metabolites of melatonin also function as free radical scavengers [2,8], improving the antioxidant quality in tissues [12,13,14,15,16,17,18,19,20,21]. Melatonin increases the SOD concentration and transforms O_2_ radicals into H_2_O_2_, which is transformed into water and oxygen by CAT and GPx. These enzymes are the first line of antioxidant defense against free-radical-induced toxicity, preventing tissue damage. Additionally, free radical scavenging actions are induced by melatonin metabolites, such as C_3_OHM, AFMK, and AMK [32].

We evaluated the antioxidant action after administering melatonin in different experimental animal models and tissues: (a) hepatoprotection (decrease in tissue damage); (b) potential action against neurotoxicity (neuronal damage); (c) improvement in renal function and potential action against nephrotoxicity; (d) improvement in male gonadotoxicity; (e) cardioprotection with lipid reduction; and (f) cytoprotection in injuries resulting from ischemia reperfusion [12,13,14,15,16,17,18,19,20,21]. These data indicate the therapeutic potential of melatonin in various conditions. These studies also suggest that its action may go beyond its antioxidant capacity, as melatonin has demonstrated other mechanisms of action in inflammatory, apoptotic, and oxidative system stability [12,13,14,15,16,17,18,19,20,21].

The anti-inflammatory effects of melatonin occur by inhibiting the transcription factors involved in generating pro-inflammatory cytokines (e.g., activator protein 1, hypoxia-inducible factor-1α, nuclear factor erythroid 2 factor 2, and nuclear factor kappa B), prostaglandin synthesis, reduced the production of adhesion molecules, neutrophil infiltration into tissues, and COX-2 (cyclooxygenase-2) levels; aiding inducible nitric oxide synthase, and immune response T-helper expression; and protecting against the cytotoxicity of heavy metals, such as lead, arsenic, chromium, cadmium, and aluminum [2,8].

Melatonin also has anti-apoptotic actions, thereby inhibiting the mitochondrial apoptotic pathway by reducing Bcl-2 (B-cell lymphoma 2) expression and caspase-3 activity [22]. This anti-apoptotic action reduces ischemia–reperfusion-induced injuries. Since necrosis can activate an immediate immune response during ischemic injury, the induction of an inflammatory reaction and the release of free radicals lead to increased oxidative stress, cell death, and organ damage [12,13,14,15,16,17,18,19,20,21,22,23,24,25,30,33,34,35,36,37,38]. This action of melatonin against apoptosis is crucial in various tissues for survival and has even been used in experimental treatments for organ transplants because it protects the tissue against ischemia and reperfusion injuries [4,8,22,23,24,25,26,27,28,29,33,34,35,36,37,38,39,40].

Melatonin may decrease MDA levels, a stable end product of lipid peroxidation, which is an indirect indicator of increased intracellular ROS generation. Lipid peroxidation is a radical chain reaction that decreases polyunsaturated fatty acids, disturbing the normal fluidity and permeability of the cell membranes and leading to cellular edema, calcium and sodium ion overload, and cell lysis. Previous studies have demonstrated reduced MDA levels by using melatonin, indicating a possible return to oxidative system stability [12].

In reproduction, melatonin improves female ovarian autotransplantation [34]. Intraperitoneal administration of melatonin caused low MDA levels, high SOD and GPx levels, and low necrosis of ovarian tissue [41,42]. To better elucidate the benefits of melatonin in reproduction, we conducted a study that demonstrated the beneficial effects of pretreatment with melatonin during cryopreservation and ovarian transplantation in an animal model. We observed an early resumption of the estrous cycle and a significant reduction in ovarian damage. The results showed (a) that the number of mature and viable follicles was greater than the vehicle; (b) an increased amount of type I and III collagen, which is essential for supporting normal ovarian structure; (c) an improved von Willebrand factor which indicated an increase in the number of vessels; and (d) decreased cleaved caspase-3, which is one of the last steps for apoptosis in the corpora lutea, and a reduced number of apoptotic bodies in the follicles and corpora lutea. These effects are possibly related to antioxidant effects [22,23,24,25,26,27,28,29,33,34,35,36,37,38,39,40,41,42]. However, more studies are needed to prove this action of melatonin in the ovaries.

In the male reproductive system, melatonin also presents several benefits. In one of the studies included in the review, treatment with potassium dichromate (PDC) decreased testosterone production owing to an increase in testicular nitric oxide or tumor necrosis factor-alpha. Furthermore, increased oxidative stress had an adverse effect on testicular function and sperm quality owing to the lower response of the antioxidant system. This study demonstrated that co-administering melatonin with PDC decreased the MDA levels and increased the SOD, CAT, and GSH activity, indicating that melatonin can reduce the changes produced by PDC in spermatogenesis by maintaining testosterone and preserving spermatogenesis and Leydig cells. These results indicate the ability of melatonin to reduce PCD-induced damage, which affects male fertility through oxidative damage [16].

Melatonin can also reduce diabetic complications, mainly by reducing lipid peroxidation. The data revealed that melatonin induced a significant reduction in MDA concentrations and oxidative damage, which is associated with decreased triglycerides and low-density lipoprotein levels and increased high-density lipoprotein levels. Therefore, melatonin could reduce diabetes-related cardiovascular diseases in an animal model [21].

The action of MDA may be tissue-dependent. A previous study analyzed the effects of melatonin combined with VPA, an anticonvulsant that negatively affects memory and hippocampal neurogenesis. Melatonin was an effective neuroprotector against these adverse effects, counteracting VPA-induced neurogenesis and memory impairment. Melatonin also reduces intracellular oxidative stress and increases the activity of SOD, CAT, and GPx; however, melatonin has a low impact on reducing MDA levels in rats treated with VPA, possibly due to the significant increase in these MDA levels in the hippocampus with the use of VPA [15].

The major limitation of these studies is demonstrate the effect of melatonin through its receptor. Therefore, a study using a melatonin receptor knockout animal model may be useful for investigating the actual melatonin mechanisms and idealizing a potential melatonin dose. The studies presented other limitations, such as (a) heterogeneity, (b) varied doses and routes of melatonin administration, (c) a lack of measurement of the plasma melatonin levels for evaluation and comparison, and (d) a lack of blinding of the evaluators. Therefore, a meta-analysis could not be performed.

The strengths of the studies included care in the treatment and housing of the animals, applying good quality measures following the local animal housing rules, maintaining lighting at the appropriate time, and simulating the light/dark cycle. However, one study did not describe these procedures [21].

Melatonin may exert protective antioxidant action in different experiments and tissues: neurological diseases, organ transplants, chronic diseases, reproduction, surgery, and cancer. Evaluating SOD, CAT, GPx and MDA is essential for understanding the beneficial effects of melatonin. These processes may be similar to those of humans. Clinical trials have shown that melatonin decreases the oxidative process in humans [43]. However, this review exclusively included studies involving animal models due to the greater availability of material in the literature to perform a more homogeneous study. These studies are essential to understanding how melatonin reduces oxidative stress and protects against tissue damage; however, further research is needed to validate these positive effects in humans.

## 5. Conclusions

Melatonin exhibits antioxidant properties, including neurological diseases, organ transplants, chronic diseases, surgery, cancer, and reproduction in several tissues and organs in different experimental models. The primary mechanism by which melatonin decreases the antioxidant process involves SOD, CAT, GPx and MDA. These benefits may be derived from their direct and indirect effects on cells. However, further research using animal models is required to elucidate the mechanisms underlying the development of new clinical applications and approaches.

## Figures and Tables

**Figure 1 antioxidants-13-00439-f001:**
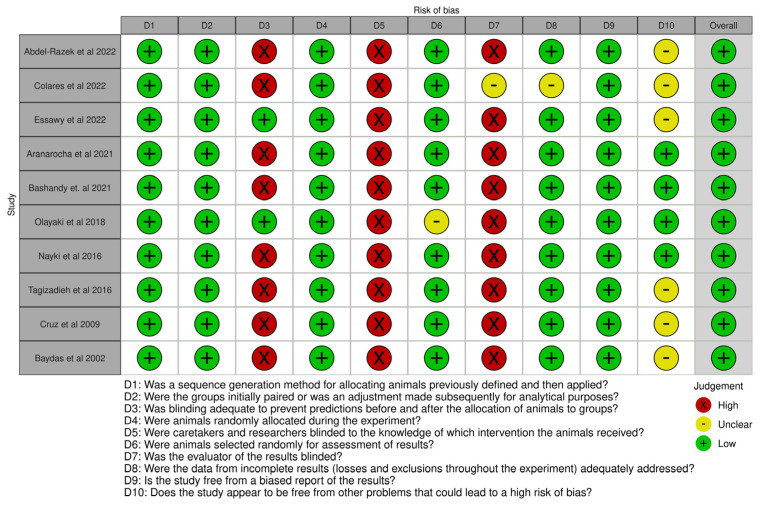
Risk of bias using the Systematic Review Center for Laboratory Animal Experiments [12,13,14,15,16,17,18,19,20,21].

**Figure 2 antioxidants-13-00439-f002:**
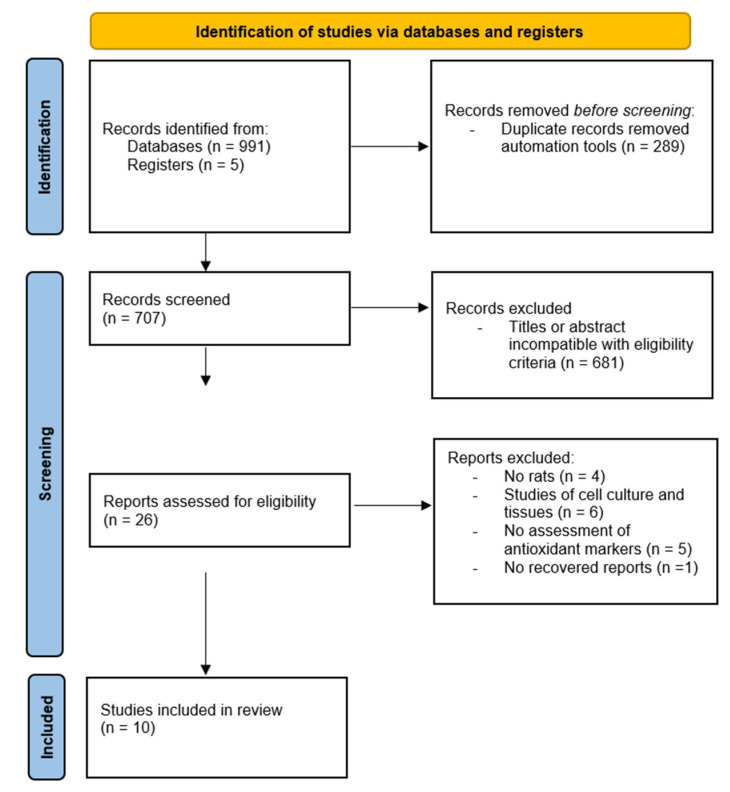
Flowchart of identification of new studies via databases and records.

**Figure 3 antioxidants-13-00439-f003:**
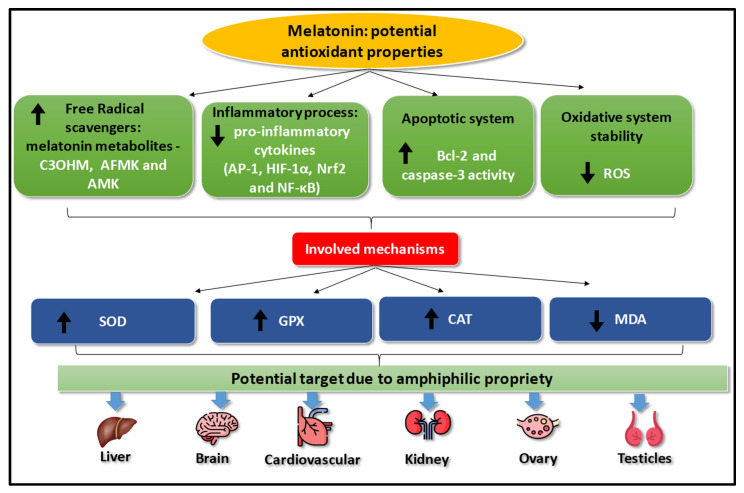
Antioxidant action of melatonin on different tissues. The potential antioxidant properties of melatonin may be related to a) the cascade antioxidant signaling independent receptor [Cyclic 3-hydroxymelatonin (C_3_OHM); N1-acetyl-N2-formyl-5-methoxykynuramine (AFMK); N1-acetyl-5-methoxykynuramine (AMK)]; b) inflammatory processes [(activator protein 1 (AP-1); hypoxia-inducible factor 1α (HIF-1α); nuclear factor erythroid 2 factor 2 (Nrf2); nuclear factor kappa B (NF-κB)]; c) the apoptotic system (Bcl-2 [B-cell lymphoma 2]); and d) oxidative system stability [reactive oxygen species (ROS)]. All these processes are related to an increase in enzymatic substances (superoxide dismutase [SOD], glutathione peroxidase [GPx], and catalase [CAT]) and a decrease in the lipid peroxidation marker malondialdehyde (MDA) as indicated by the arrows in the figure. Melatonin is an amphiphilic molecule (hydrophilic and lipophilic) that may be capable of diffusing into different tissues. The potential targets of melatonin are the a) liver, brain, cardiovascular system, kidneys, ovaries, and testicles [12,13,14,15,16,17,18,19,20,21].

**Table 1 antioxidants-13-00439-t001:** Medical Subject Headings terms used in our database search.

MeSH Terms
Oxidative Stresses Stress OR Oxidative Antioxidative OR Stress Antioxidative Stresses Stress OR Antioxidative OR Antioxidative Stress OR Antioxidative Stress OR Antioxidative Stresses Stress OR Antioxidative Oxidative Damage OR Oxidative Damages OR Oxidative Stress Injury OR Oxidative Stress Injuries OR Stress Injury OR Oxidative Injury OR Oxidative Cleavage OR Oxidative DNA Damage OR DNA Oxidative and Nitrosative Stress OR Nitrative Stress AND (rats) AND (glutathione) AND (GPx) AND (malondialdehyde) AND (SOD) AND (superoxide dismutase) AND (catalase) AND (CAT)
Oxidative Stresses Stress OR Oxidative Antioxidative OR Stress Antioxidative Stresses Stress OR Antioxidative OR Antioxidative Stress OR Antioxidative Stress OR Antioxidative Stresses Stress OR Antioxidative Oxidative Damage OR Oxidative Damages OR Oxidative Stress Injury OR Oxidative Stress Injuries OR Stress Injury OR Oxidative Injury OR Oxidative Cleavage OR Oxidative DNA Damage OR DNA Oxidative and Nitrosative Stress Oxidative Nitrative Stress OR Nitrative Stress OR Oxidative Nitrative Stresses Stress OR Nitro-Oxidative Stress OR Nitro Oxidative Stress OR Nitro-Oxidative Stresses Stress, Nitro-Oxidative Stresses OR Nitro-Oxidative) AND (melatonin OR Indoles OR Tryptamines) AND (rats) AND (mice)
Oxidative Stresses Stress OR Oxidative Antioxidative OR Stress Antioxidative Stresses Stress OR Antioxidative OR Antioxidative Stress OR Antioxidative Stress OR Antioxidative Stresses Stress OR Antioxidative Oxidative Damage OR Oxidative Damages OR Oxidative Stress Injury OR Oxidative Stress Injuries OR Stress Injury OR Oxidative Injury OR Oxidative Cleavage OR Oxidative DNA Damage OR DNA Oxidative and Nitrosative Stress Oxidative Nitrative Stress OR Nitrative Stress OR Oxidative Nitrative Stresses Stress OR Nitro-Oxidative Stress OR Nitro Oxidative Stress OR Nitro-Oxidative Stresses Stress, Nitro-Oxidative Stresses OR Nitro-Oxidative) AND (melatonin OR Indoles OR Tryptamines) AND (rats) AND (glutathione)
(Oxidative Stress OR Oxidative Antioxidative) AND (melatonin OR Indoles OR Tryptamines) AND (rats) AND (glutathione) AND (GPx) AND (malondialdehyde) AND (SOD) AND (superoxide dismutase) AND (catalase) AND (CAT)
(Melatonin OR Indoles OR Tryptamines) AND (glutathione) AND (glutathione peroxidase) AND (malondialdehyde) AND (SOD) AND (superoxide dismutase) AND (catalase) AND (CAT)

**Table 2 antioxidants-13-00439-t002:** Melatonin doses, route of administration, and results.

Authors	No. of Rats	Melatonin Dose	Tests	Results
Abdel-Razeket al. 2022 [12]	48	10 mg/kg/d Intraperitoneal route	MDA SOD GPx	decreased ++ increased ++ increased +
Colares et al. 2022 [13]	24	20 mg/kg/d	SOD	increased ++
Essawy et al. 2022 [14]	40	10 mg/kg/d Oral route	MDA SOD GPx CAT	decreased ++ increased ++ increased ++ increased ++
Aranarocha et al. 2021 [15]	48	8 mg/kg/d Intraperitoneal route	SOD GPx CAT MDA	increased ++ increased ++ increased ++ no difference
Bashandy et al. 2021 [16]	40	2.5 or 5 mg/kg/d Intraperitoneal route	MDA SOD CAT GSH	decreased ++ increased ++ increased ++ increased ++
Olayaki et al. 2018 [17]	50	4 or 10 mg/kg/d Oral route	MDA SOD GPx GSH	decreased ++ increased ++ increased ++ increased ++
Nayki et al. 2016 [18]	24	20 mg/kg/d Intraperitoneal route	MDA SOD CAT TOS TAS	decreased ++ increased ++ increased + decreased ++ increased ++
Tagizadieh et al. 2016 [19]	60	20 mg/kg/d Intraperitoneal route	GPx SOD MDA	increased ++ increased ++ decreased ++
Cruz et al. 2009 [20]	21	0.5 mg /kg/d Intraperitoneal route	MDA GSH CAT SOD GPx	decreased ++ increased ++ increased ++ increased ++ increased ++
Baydas et al. 2002 [21]	80	0.1 mg/kg/d Intraperitoneal route	GPx	increased ++

Malondialdehyde (MDA), superoxide dismutase (SOD), glutathione peroxidase (GPx), reduced glutathione (GSH), and catalase (CAT); effect increased/decreased + slightly/++ strongly.

## Data Availability

All the data and materials are included in the article.

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
