# Peer review of "Antioxidant Actions of Melatonin: A Systematic Review of Animal Studies"

_antioxidants, 2024, doi:10.3390/antiox13040439_

Round 1

Reviewer 1 Report

Comments and Suggestions for Authors

In this manuscript, the authors conducted a systematic review of antioxidant properties of melatonin in animal models. However, this paper does not make a significant contribution to the scientific literature. It lacks novelty and is not innovative.

As a suggestion. The authors should include in the title “a systematic review of animal studies”.

They mention that melatonin has only been studied in animal models when a systematic review and meta-analysis has been previously carried out analyzing melatonin supplementation on oxidative stress in randomized controlled trials:

Ghorbaninejad et al. (2020) Effects of melatonin supplementation on oxidative stress: a systematic review and meta-analysis of randomized controlled trials. Horm Mol Biol Clin Investig 41(4). doi: 10.1515/hmbci-2020-0030.

Comments on the Quality of English Language

From my view, the scientific work described here is not suitable for publication in Antioxidants. Perhaps, other Journal with lower impact factor would be more appropriate.

Author Response

Dear Reviewer

Comments 1: The authors should include in the title “a systematic review of animal studies”.
Response 1: Thank you for pointing this out. We agree with this comment. Therefore, we included "a systematic review of animal studies" on the title.

Comments 2: They mention that melatonin has only been studied in animal models when a systematic review and meta-analysis has been previously carried out analyzing melatonin supplementation on oxidative stress in randomized controlled trials:
Ghorbaninejad et al. (2020) Effects of melatonin supplementation on oxidative stress: a systematic review and meta-analysis of randomized controlled trials. Horm Mol Biol Clin Investig 41(4). doi: 10.1515/hmbci-2020-0030.

Response 2: Thank you for pointing this out. We updated text in the manuscript.
We chose to only include studies in animal models because there is greater availability of data in studies on oxidative stress and the use of melatonin.

Please see the manuscript attachment.

Reviewer 2 Report

Comments and Suggestions for Authors

Melatonin is a very promising compound to treat numerous pathologies. Its deficiency is considered to participate in the etiology of civilization diseases. A lot of studies included melatonin supplementation, but the results are still not translated to the clinical use. Thus, the review seems to be interesting. However, in my opinion, the review should be significantly corrected, especially the articles should be better discussed. The details are below.  

. All abbreviations should be explained, while first mentioned (e.g., lines 49-50, 95 etc), and then they should be used in the whole manuscript. The abbreviations in the table/figure titles/legends should be explained independently on the main text.

2. Line 82-  The term nitrosative stress should be also mentioned.

3. Line 93 - the synthesis of glutathione is induced by melatonin, but GSH is not an enzyme, like is suggested in the sentence.

4. In the material and methods, you should explain what time period you took into account when looking for articles. 

5. English is quite fine, but it is worth improving the language with a professional.

6. Line 206 - what do you mean by the expression "increase less"?

7. And again - you mentioned several times that GSH and MDA belong to enzymes, this should be corrected in the whole manuscript.

8. I do not understand how you performed the order of the articles, especially in the Table 3, maybe you could make some order, for example, with an increased/decreased dose of melatonin and the route of administration?

9. First paragraph of the discussion repeats the information from the introduction, please, rewrite it.

10. Line 292 - the first sentence has no reference to the previous paragraph. Please, correct.

11. Generally, Discussion seems to be chaotic, is written in worse English with numerous typos. It is worth rewriting it, describing the results of the articles in more details. The presented articles should be discussed with other studies, human and in vitro studies could be mentioned.

12. In Table 3, maybe you could add the percentage of change of  enzymatic activities, not only "increase" or "decrease".

13. Line 358: in the case of enzymes, levels or activities were measured? This should be checked and corrected.

14. Line 367: what your findings do you mean?

15. Line 399: You described only animal studies, because it was your choice. There exist studies on humans, of course, more clinical studies are required, but the sentence should be rewritten.

16. It is worth to add a figure with potential melatonin antioxidant functions on the enzymatic activity or the described diseases/tissues.

17. Conclusions are to short for me. They should better describe the content of the review

18. 20 references seem to be not enough for me to discuss the presented problem, more references are required for the introduction, but also for the discussion.

Comments on the Quality of English Language

English is fine in the first chapters, Discussion has numerous mistakes and typos. I advice to correct English by a professional.

Author Response

Dear Reviewer

Thank you very much for taking the time to review this manuscript. Please find the detailed responses below and the corresponding revisions/corrections highlighted in track changes in the re-submitted files.

Comments 1: All abbreviations should be explained, while first mentioned (e.g., lines 49-50, 95 etc), and then they should be used in the whole manuscript. The abbreviations in the table/figure titles/legends should be explained independently on the main text.

Response 1: Thank you for pointing this out. We updated text in the manuscript.

Comments 2:Line 82- The term nitrosative stress should be also mentioned.

Response 2: Thank you for pointing this out. We updated text in the manuscript.

Comments 3:Line 93 - the synthesis of glutathione is induced by melatonin, but GSH is not an enzyme, like is suggested in the sentence.

Response 3: Thank you for pointing this out. We updated text in the manuscript.

Comments 4:In the material and methods, you should explain what time period you took into account when looking for articles. 

Response 4: Thank you for pointing this out. We updated text in the manuscript.

Comments 5:English is quite fine, but it is worth improving the language with a professional.

Response 5: Thank you for pointing this out. We updated text in the manuscript, we correct it with a professional.

Comments 6: Line 206 - what do you mean by the expression "increase less"?

Response 6: Thank you for pointing this out. We updated text in the manuscript.

Comments 7: And again - you mentioned several times that GSH and MDA belong to enzymes, this should be corrected in the whole manuscript.

Response 7: Thank you for pointing this out. We updated text in the manuscript.

Comments 8: I do not understand how you performed the order of the articles, especially in the Table 3, maybe you could make some order, for example, with an increased/decreased dose of melatonin and the route of administration?

Response 8: Thank you for pointing this out. We updated text in the manuscript, the order was year of publication.

Comments 9: First paragraph of the discussion repeats the information from the introduction, please, rewrite it.

Response 9: Thank you for pointing this out. We updated text in the manuscript.

Comments 10:Line 292 - the first sentence has no reference to the previous paragraph. Please, correct.

Response 10: Thank you for pointing this out. We updated text in the manuscript.

Comments 11: Generally, Discussion seems to be chaotic, is written in worse English with numerous typos. It is worth rewriting it, describing the results of the articles in more details. The presented articles should be discussed with other studies, human and in vitro studies could be mentioned.

Response 11: Thank you for pointing this out. We updated text in the manuscript.

Comments 12: In Table 3, maybe you could add the percentage of change of  enzymatic activities, not only "increase" or "decrease".

Response 12: Thank you for pointing this out. We updated it in the table in the manuscript. 

Comments 13: Line 358: in the case of enzymes, levels or activities were measured? This should be checked and corrected.

Response 13: Thank you for pointing this out. We updated text in the manuscript.

Comments 14: Line 367: what your findings do you mean?

Response 14: Thank you for pointing this out. We updated text in the manuscript.

Comments 15: Line 399: You described only animal studies, because it was your choice. There exist studies on humans, of course, more clinical studies are required, but the sentence should be rewritten.

Response 15:Thank you for pointing this out. We updated text in the manuscript.

Comments 16: It is worth to add a figure with potential melatonin antioxidant functions on the enzymatic activity or the described diseases/tissues.

Response 16: Thank you for pointing this out. We add a figure with melatonin antioxidant functions in the described diseases/tissues. Line 421.

Comments 17: Conclusions are to short for me. They should better describe the content of the review.

Response 17: Thank you for pointing this out. We updated text in the manuscript.

Comments 18: 20 references seem to be not enough for me to discuss the presented problem, more references are required for the introduction, but also for the discussion.

Response 18: Thank you for pointing this out. We updated text in the manuscript.

Please see the manuscript attachment.

Reviewer 3 Report

Comments and Suggestions for Authors

In this systematic review, the authors investigated the antioxidant properties of melatonin treatment in animal models. Following the PRISMA guidelines, the selected 10 articles addressed the melatonin effect on antioxidant markers, such as SOD, CAT, MDA, and so on. The overall results then suggested that melatonin is effective in reducing oxidative stress.

The review is interesting; here are some comments that may improve its quality:

- Line 74: What results are you referring to

- Line 79: more details on melatonin metabolites should be provided;

- Lines 82-86 need references;

- Line 119: why just studies using rats as experimental models were included, considering the potential clinical application of melatonin?

- The used mesh terms seem a little limited (for example, many papers may not have highlighted in the title or the keywords those terms, excluding papers that may have been included in the review). For this, the authors should better justify their choice.

Author Response

Dear Reviewer

Thank you very much for taking the time to review this manuscript. Please find the detailed responses below and the corresponding revisions/corrections highlighted in track changes in the re-submitted files.

Comments 1: Line 74: What results are you referring to.

Response 1: Thank you for pointing this out. We updated text in the manuscript.

Comments 2: Line 79: more details on melatonin metabolites should be provided.

Response 2: Thank you for pointing this out. We updated text in the manuscript.

Comments 3: Lines 82-86 need references.

Response 3: Thank you for pointing this out. We updated text in the manuscript.

Comments 4: Line 119: why just studies using rats as experimental models were included, considering the potential clinical application of melatonin?

The used mesh terms seem a little limited (for example, many papers may not have highlighted in the title or the keywords those terms, excluding papers that may have been included in the review). For this, the authors should better justify their choice.

Response 4: Thank you for pointing this out. We updated text in the manuscript.

Please see the manuscript attachment.

Round 2

Reviewer 1 Report

Comments and Suggestions for Authors

The authors have answered to all my concerns

Author Response

Dear Reviewer #1

The authors have answered to all my concerns

Thank you for your comment.

Reviewer 2 Report

Comments and Suggestions for Authors

I have read the revised version and now have greater doubts as to whether the article is of sufficient quality to be published in the journal.

1. There are numerous English mistakes/typos in the revised part (starting with line 33 - should be "recent years", line 47 - should be "has been shown", line 78 - the sentence must not start with "Which"). 

2. Lines 148-149: it is still "the following enzymes as antioxidant action markers: MDA, SOD, GPx, GSH and CAT" - thus, you have not revised the manuscript according to my comments. It is not enough to answer "we updated the text" - the text should be revised in reality.

3. The Figure 2 is not convincing to me. I expected a figure that would present the mechanisms mentioned in the discussion in more detail, but in its current form it does not provide anything interesting or summarizing.

4. Lines 284-287: the sentence "The, evidence of the participation of melatonin is not clear, since melatonin molecules are amphiphilic, presenting hydrophilic and lipophilic properties, which might influence, Vitamin C (hydrophilic) and E (lipophilic) in various subcellular compartments, 286 such as membranes, cytoplasm, nucleus, and mitochondria, expanding their action".  For me, the sentence does not make much sense. It is one of numerous examples.

5. Discussion seems to be more chaotic than before the revision.

Comments on the Quality of English Language

I cannot see the article has been revised by an English professional. Grammar mistakes and typos are common in the text.

Author Response

Re: Resubmission of manuscript “Antioxidant actions of melatonin: a systematic review of animal studies"

We would like to resubmit our manuscript entitled “Antioxidant actions of melatonin: a systematic review of animal studies" after suggestions of reviewers.

Thank you very much for taking the time to review this manuscript. Please find the detailed responses below and the corresponding revisions/corrections highlighted/in track changes in the re-submitted files. All author agree with this version of manuscript (highlighted changes) and the response of all peer reviewer are below:

Comments 1: There are numerous English mistakes/typos in the revised part (starting with line 33 - should be "recent years", line 47 - should be "has been shown", line 78 - the sentence must not start with "Which").

Response 1: Thank you for pointing this out. We changed text in the manuscript.

Comments 2: Lines 148-149: it is still "the following enzymes as antioxidant action markers: MDA, SOD, GPx, GSH and CAT" - thus, you have not revised the manuscript according to my comments. It is not enough to answer "we updated the text" - the text should be revised in reality.

Response 2: Thank you for pointing this out. We changed text in the manuscript.

Comments 3: The Figure 2 is not convincing to me. I expected a figure that would present the mechanisms mentioned in the discussion in more detail, but in its current form it does not provide anything interesting or summarizing.

Response 3: Thank you for pointing this out. We changed text in the manuscript.

Comments 4: Lines 284-287: the sentence "The, evidence of the participation of melatonin is not clear, since melatonin molecules are amphiphilic, presenting hydrophilic and lipophilic properties, which might influence, Vitamin C (hydrophilic) and E (lipophilic) in various subcellular compartments, 286 such as membranes, cytoplasm, nucleus, and mitochondria, expanding their action".  For me, the sentence does not make much sense. It is one of numerous examples.

Response 4: Thank you for pointing this out. We updated text in the manuscript.

Comments 5: Discussion seems to be more chaotic than before the revision.

Response 5: Thank you for pointing this out. We changed the discuss in the manuscript.